# QA-NatVer: Question Answering for Natural Logic-based Fact Verification

**Rami Aly, Marek Strong, Andreas Vlachos**
Department of Computer Science and Technology
University of Cambridge
{rmya2,ms2518,av308}@cam.ac.uk

## Abstract

Fact verification systems assess a claim's veracity based on evidence. An important consideration in designing them is faithfulness, i.e. generating explanations that accurately reflect the reasoning of the model. Recent works have focused on natural logic, which operates directly on natural language by capturing the semantic relation of spans between an aligned claim with its evidence via set-theoretic operators. However, these approaches rely on substantial resources for training, which are only available for high-resource languages. To this end, we propose to use question answering to predict natural logic operators, taking advantage of the generalization capabilities of instruction-tuned language models. Thus, we obviate the need for annotated training data while still relying on a deterministic inference system. In a few-shot setting on FEVER, our approach outperforms the best baseline by 4.3 accuracy points, including a state-of-the-art pre-trained seq2seq natural logic system, as well as a state-of-the-art prompt-based classifier. Our system demonstrates its robustness and portability, achieving competitive performance on a counterfactual dataset and surpassing all approaches without further annotation on a Danish verification dataset. A human evaluation indicates that our approach produces more plausible proofs with fewer erroneous natural logic operators than previous natural logic-based systems.

## 1 Introduction

Automated fact verification is concerned with the task of identifying whether a factual statement is true, with the goal of improving digital literacy (Vlachos and Riedel, 2014). A typical fact verification system consists of an evidence retrieval and a claim verification component (i.e. the judgement of whether a claim is true). The latter is typically implemented as a neural entailment system (Guo et al., 2022, *inter alia*), which is not transparent in regards to its underlying reasoning. While efforts have been made to improve their explainability, for instance via highlighting salient parts of the evidence (Popat et al., 2018), or generating summaries (Kotonya and Toni, 2020), there is no guarantee that the explanations are *faithful*, i.e. that they accurately reflect the reasoning of the model (Jacovi and Goldberg, 2020; Atanasova et al., 2023).

Contrarily, proof systems like NaturalLI (Angeli and Manning, 2014), perform natural logic inference as proofs, and are faithful by design. Their transparent reasoning empowers actors to make informed decisions about whether to trust the model and which parts of the prediction to dispute (Jacovi and Goldberg, 2021). Recently Krishna et al. (2022) constructed a natural logic theorem prover for claim verification, using an autoregressive formulation with constrained decoding. However, an important limitation of this approach is its dependence on substantial resources for training, relying on large datasets for claim verification, and structured knowledge bases like the ParaPhrase DataBase (Ganitkevitch et al., 2013), WordNet (Miller, 1994), and Wikidata (Vrandečić and Krötzsch, 2014). However such manually curated resources are typically accessible for high-resource languages, thus limiting its applicability.

To this end, we propose **QA-NatVer**: **Q**uestion **A**nswering for **Nat**ural Logic-based Fact **Veri**fication, a natural logic inference system that composes a proof by casting natural logic into a question answering framework. As illustrated in Figure 1, a proof is a sequence of steps, with each step describing the semantic relation between a claim span and an evidence span via a set-theoretic natural logic operator (NatOp), and this sequence of NatOps determines the veracity of the claim following a deterministic finite state automaton (DFA). QA-NatVer predicts NatOps using operator-specific boolean questions (cf. Table 1 for examples for all operators). For instance, the relation between the claim span

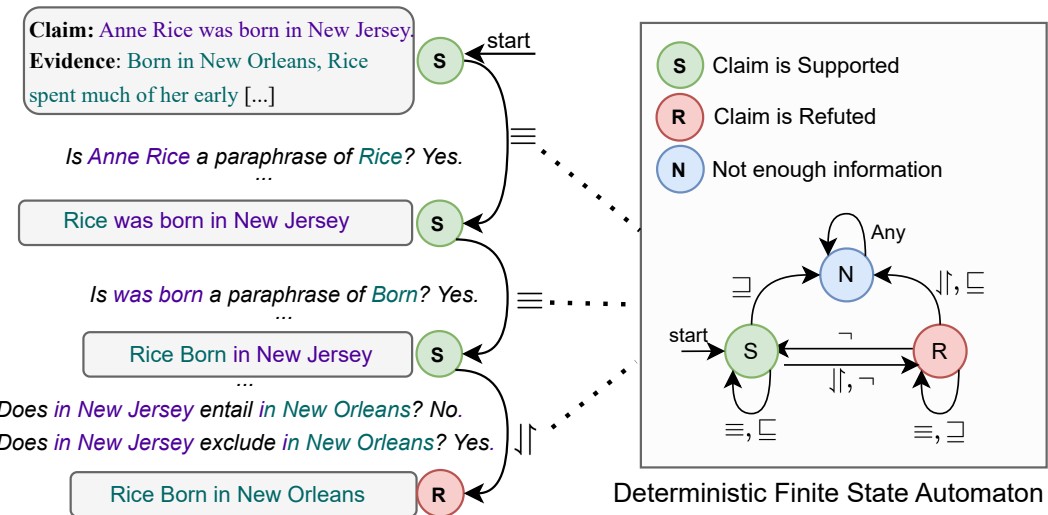

Figure 1: At each inference step, a claim span is mutated into an evidence span via a natural logic operator (NatOp). The current veracity state and mutation operator determine the transition to the next state, via a fine state automaton (DFA). Starting at **S**, the span *Anne Rice* is mutated via the equivalence operation (≡), resulting in **S**, according to the DFA. The inference ends in **R**, indicating the claim's refutation. We use question-answering to predict the NatOps, taking advantage of the generalization capabilities of instruction-tuned language models.

*was born* and the evidence *Born* is ascribed the equivalence NatOp (≡), which we predict with questions such as *Is "was born" a paraphrase of "Born"?*. This formulation enables us to make use of instruction-finetuned language models, which are powerful learners, even with limited supervision (Sanh et al., 2022, *inter alia*). Since the input format to our question-answering formulation constrains the context to the aligned claim-evidence spans, we generate claim-evidence alignments between overlapping spans of varying length, and individually predict the NatOp for each pair of aligned spans. To select the best proof over all possible proofs, we combine the answer scores to the questions associated with each proof.

In a few shot setting with 32 training instances on FEVER, QA-NatVer outperforms all baselines by 4.3 accuracy points, including ProofVER, LOREN (Chen et al., 2022a), and a state-of-the-art few-shot classification system, T-Few(Liu et al., 2022). By scaling the instruction-tuning model from BART0 to Flan-T5, we achieve a score of $70.3 \pm 2.1$, closing the gap to fully supervised models, trained on over 140,000 instances, to $8.2$ points. On an adversarial claim verification dataset, Symmetric FEVER (Schuster et al., 2019), QA-NatVER scores higher than few-shot and fully supervised baselines, including ProofVER, demonstrating the robustness of our approach. In a low-resource sce-

nario on a Danish fact verification dataset (Nørregaard and Derczynski, 2021, DanFEVER) without further annotation for training, our system outperforms all baselines by $1.8$ accuracy points, highlighting the potential of the question-answering formulation for low-resource languages. An ablation study indicates the benefits of QA-NatVer's question-answering formulation, outperforming ChatGPT (OpenAI, 2022) (over 430x the size of BART0) prompted with in-context examples to predict NatOps by $11.9$ accuracy points. Finally, we show in a human evaluation that QA-NatVer improves over previous natural logic inference systems in terms of explainability, producing more plausible proofs with fewer erroneous NatOps.[1]

## 2 Related Work

Natural logic (Van Benthem, 1986; Sanchez, 1991) operates directly on natural language, making it an appealing alternative to explicit meaning representations such as lambda calculus, since the translation of claims and evidence into such representations is error-prone and difficult to decode for non-experts. NatLog (MacCartney and Manning, 2007, 2009) proposes the use of natural logic for textual inference, which has then subsequently been extended by Angeli and Manning (2014) into the Nat-

---

[1]Code and models are available at https://github.com/Raldir/QA-NatVer

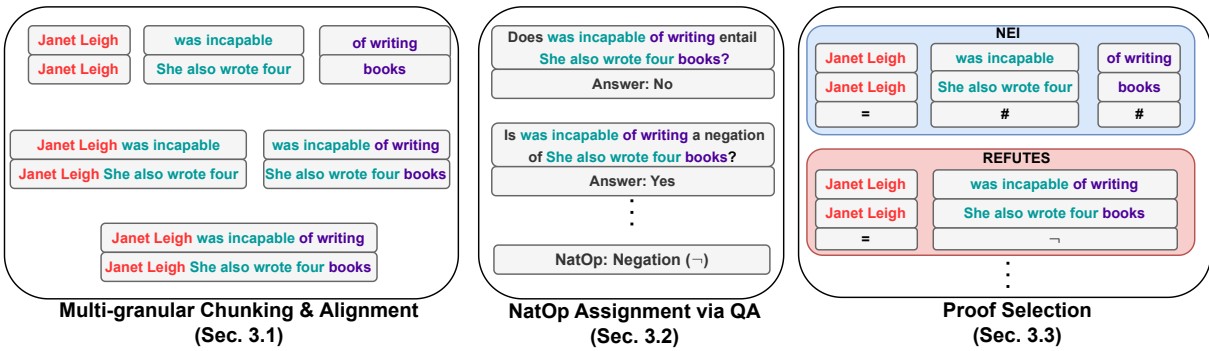

Figure 2: QA-NatVer's proof construction. We first chunk the claim and the evidence, and align them at multiple granularity levels. We then assign a natural logic operator to each aligned claim-evidence span using question-answering. Finally, we select the proof by combining the answer scores to the questions associated with the proof.

uralLI proof system. With the surge of pre-trained language models, multiple works have attempted to integrate natural logic into neuro-symbolic reasoning systems (Feng et al., 2020, 2022). In particular, ProoFVer, a natural logic inference system specifically designed for fact verification, achieves competitive performance yet remains faithful and more explainable than its entirely neural approaches (Krishna et al., 2022). Stacey et al. (2022) propose an alternative framework of logical reasoning, evaluating the veracity of individual claim spans (or of atomic facts in Stacey et al. (2023)) and determining the overall truthfulness by following a simple list of logical rules. Chen et al. (2022a) use a similar list of logical rules but aggregate the outcomes with a neural network component. However, all previous approaches have in common that they require substantial training data to perform well, limiting their use to resource-rich languages and domains.

Casting a natural language problem to a question-answering setting has previously been explored in a variety of tasks such as relation classification (Levy et al., 2017; Cohen et al., 2022), and semantic role labeling (He et al., 2015; Klein et al., 2022). For fact verification, in particular, previous works have considered formulating it as a question generation task, decomposing a claim into relevant units of information to inquire about, followed by question answering to find relevant answers in a large knowledge base (Fan et al., 2020; Chen et al., 2022b). Yet, these works do not consider aggregating nor constructing proofs from the answers to the questions.

Finally, work on few-shot claim verification is limited. Lee et al. (2021) explore using a perplexity score, however, their approach is constrained to

binary entailment, i.e. either supported or refuted. Zeng and Zubiaga (2023) explore active learning in combination with PET (Schick and Schütze, 2021), a popular prompt-based few-shot learning method, and Pan et al. (2021) and Wright et al. (2022) generate weakly supervised training data for zero-shot claim verification. However, none of the aforementioned methods produces (faithful) explanations.

## 3 Method

Given a claim $c$ and a set of $k$ evidence sentences $E$, the task of claim verification is to predict a veracity label $\hat{y}$ and to generate a justification for the selected label. QA-NatVer is a system that returns a natural logic inference proof which consists of a sequence of relations $P = m_1, \dots, m_l$, each of them specifying a relation between a claim and evidence span and a NatOp operator $o$.[2] The sequence of operators $O = o_1, \dots, o_l$ is then the input to a deterministic finite state automaton that specifies the veracity label $\hat{y} = \text{DFA}(O)$ (c.f. Figure 1). The proof $P$ itself serves as the justification for the predicted label $\hat{y}$.

QA-NatVer constructs its proofs following a three-step pipeline, illustrated in Figure 2: multi-granular chunking of the claim and its evidence sentences and the alignment between claim-evidence spans (Sec. 3.1), assignment of NatOps to each aligned pair using question answering (Sec. 3.2), and a proof selection mechanism over all possible proofs by combining the answer probabilities to the questions associated with the proof (Sec. 3.3).

---

[2]In the natural logic literature these operators define mutations that would convert the claim so that it follows from the evidence (Angeli and Manning, 2014), however we do not make use of the edited claim in this paper.

## 3.1 Multi-granular Chunking & Alignment

We chunk the claim initially into $l$ non-overlapping consecutive spans $c = c_1, \ldots, c_l$, using the chunker of Akbik et al. (2019), and merge spans that do not contain any content words with their subsequent spans. To align each claim span $c_i$ with the information of the highest semantic relevance in the evidence $E$, we use the fine-tuned contextualized word alignment system of Dou and Neubig (2021) to first align individual words between the claim and each evidence sentence $E_j$. These word-level alignments are then mapped back to the span $c_i$ to form an evidence span $e_{ij}$ from the sentence $E_j$. Since multiple spans in the evidence sentences could align with $c_i$, we measure the cosine similarity between $c_i$ and each aligned evidence span $e_{ij}$, using latent span embeddings via Sentence Transformers (Reimers and Gurevych, 2019).

It is of essence that the granularity of a claim's span matches the evidence span to capture their semantic relationship correctly. Therefore, we additionally consider merging the claim chunks $c_1, \ldots, c_l$ into more coarse-grained chunks. Concretely, we concatenate $m$ consecutive claim chunks into a single new span $c_{i:i+m}$, with $m$ being up to a length of $4$. The merging process results in a total of at most $q = 4 \cdot l - 6$ chunks. Additionally, we consider the claim $c$ itself as the most coarse-grained unit and align it to evidence in $E$.

Consider the example in Figure 2. A system that only considers a single chunking might consider *was incapable* and *of writing* as separate phrases. However, the evidence spans aligned to these individual phrases (*She also wrote four* and *books*, respectively) do not provide enough context individually to infer their semantic relations with respect to the claim spans. However, when merged into a single chunk, their semantic relation becomes obvious, as the span *was incapable of writing* is negated by *she also wrote four books*. Hence, a more flexible variable-length chunking enables finding more semantically coherent alignments.

## 3.2 NatOp Assignment via QA

Each claim-evidence pair has to be assigned a NatOp, specifying their semantic relationship. We assign one out of six NatOps $o \in \{\equiv, \sqsubseteq, \sqsupseteq, \neg, \downarrow\upharpoonright, \#\}$[3] to each claim-evidence span. We formu-

late the prediction for each NatOp $o$ as a question-answering task (cf. Table 1), each of which is instantiated by one or more boolean question prompts $T_o$. Only exception is the independence operator (#) which is applied when none of the other operators are predicted. To predict whether a NatOp $o$ holds between a claim span $c_i$ aligned with evidence $e_i$, we compute the log probabilities averaged over all question prompts $T_o$:

$$\text{QA}(a \mid c_i, e_i, T_o) = \frac{1}{|T|} \sum_{t \in T_o} \log p_\theta(a \mid c_i, e_i, t), \tag{1}$$

with $a \in \{\text{Yes}, \text{No}\}$, $|T|$ being the number of question prompts, and QA being our seq2seq instruction-tuned language model (see App. A for details). We apply an argmax function to select the most probable answer $\hat{a}_o = \text{argmax}_y \text{QA}(a \mid c_i, e_i, T_o)$. An answer prediction $\hat{a}_o = $ *Yes* indicates that the NatOp $o$ holds for the aligned claim-evidence spans. This formulation enables us to effectively use of the generalization abilities of instruction-tuned language models.

As illustrated in Figure 2, given the aligned claim-evidence spans *was incapable of writing* and *She also wrote four books*, we ask questions for each of the five NatOps, with the spans embedded in them. In the figure, the negation NatOp ($\neg$) is selected due to its corresponding boolean question being positively answered. Since predictions are made independently for each NatOp, it can occur that the model predicts multiple NatOps for a single pair of aligned claim-evidence spans. In these instances, we select the NatOp with the highest probability (as computed in Eq. 1). On the contrary, if none of the five NatOps is predicted, we assign the independence operator (#).

| NatOp | Task | Question Example |
|---|---|---|
| Equivalence ($\equiv$) | Paraphrase identification | Is in New Jersey a paraphrase of in New Orleans? |
| Fwrd. Entailment ($\sqsubseteq$) | Entailment | Given the premise in New Orleans does the hypothesis in New Jersey hold? |
| Rev. Entailment ($\sqsupseteq$) | Entailment | Does in New Jersey entail in New Orleans? |
| Negation ($\neg$) | Negation classification | Is the phrase in New Jersey a negation of in New Orleans? |
| Alternation ($\downarrow\upharpoonright$) | Alternation classification | Does in New Jersey exclude in New Orleans? |

Table 1: Natural logic operators and the corresponding natural language task and boolean question examples.

---

[3]Similarly to Angeli and Manning (2014) and Krishna et al. (2022), we do not consider the rarely appearing cover operator, but instead replace it with the independence NatOp.

### 3.3 Proof Selection

Since we expand the $l$ initial non-overlapping claim chunks with multi-granular merging into $q$ overlapping ones, we can construct a total of $C(l) = \sum_{i=l-m}^{l-1} C(i)$ proofs, with $C(i)$ being the number of proofs for $i$ chunks, $C(0) = 1$, and $m$ being the maximum merge length.[4] To select the most appropriate proof we compute a score for each one, defined as the sum of a NatOp probability score $s_p$ (Eq. 2) and a NatOp verdict score $s_v$ (Eq. 3) introduced later in this section. We select the proof with the highest score.

Since the probability for each NatOp is computed independently, we define the score $s_p$ as the average of the predicted NatOp probabilities:

$$s_p = \frac{1}{n} \sum_{i=1}^{n} \text{QA}(a_{\text{Yes}} \mid c_i, e_i, T_o), \qquad (2)$$

with $n$ being the length of the proof $P$, and $T_o$ being the questions for the NatOp $o$ assigned to the $i$-th aligned span in the proof. Since no probability is explicitly assigned to the independence operator (#) as we have no question to directly capture it (cf. Section 3.2), we use the lowest scoring option to be added to the average in these cases (i.e. $0.5$ since the predictions are boolean).

The NatOp verdict score $s_v$ considers the aligned claim with its evidence in its entirety as the most coarse-grained chunk, for which our question-answering system computes a score over a textual representation of the veracity labels $y$:

$$s_v = \text{QA}(y_{\text{DFA}(O)} \mid c, E, T_v), \qquad (3)$$

with $T_v$ being the veracity question templates, and $O$ being the NatOp sequence in the proof $P$. The score $s_v$ is defined to be the probability assigned to the veracity label associated with the state in the DFA in which the proof $P$ would terminate in, i.e. DFA(O). In our example, the proof where *was incapable* and *of writing* are considered separate spans receive a low score due to the two independence NatOps and its termination in the *Not enough info* (NEI) state, while the one where they are merged in a single span is selected due to high answer confidence in the predicted NatOps.

---

[4]While the number of potential proofs can grow large, computations for each claim-evidence span are only made once, resulting in linear complexity with respect to $l$, c.f. Section 4.5.

### 3.4 Training

We fine-tune the word-level alignment system as well as our question-answering system for QA-NatVer. The alignment system is trained on multiple training objectives as defined in Dou and Neubig (2021) for parallel corpora, namely masked language modelling, translation language modelling, and their self-training objective. We consider the claim $c$ and each gold evidence $e \in E_G$ as a sentence pair. We create further pairs using gold proofs $P_G$ by considering all possible substitutions of claim chunks with their respective evidence chunk. Our question-answering system is fine-tuned following Liu et al. (2022), by optimizing the maximum likelihood estimation (cf. Eq. 1), complemented by an unlikelihood loss which discourages the model from predicting incorrect target sequences. The NatOps in the gold proofs $P_G$ are used as positive QA samples, while we sample negative training instances (i.e. NatOp questions with the answer being "No") from the gold proofs by randomly selecting a wrong NatOp for an aligned claim-evidence span.

## 4 Evaluation

### 4.1 Few-Shot Claim Verification

**Data** We manually annotated 68 training instances of FEVER (Thorne et al., 2018) with natural logic proofs. Since FEVER does not contain gold evidence for instances labeled with NEI, we use retrieved evidence instead. The samples were drawn to maintain a balanced label distribution, with 22 Supported, 25 Refuted, and 21 NEI instances. This results in proofs with a total of 183 Equivalence ($\equiv$), 55 Forward Entailment ($\sqsubseteq$), 16 Reverse Entailment ($\sqsupseteq$), 29 Negation ($\neg$), 31 Alternation ($|$) , and 40 independence (#) NatOps. We train all systems on 32 samples unless stated otherwise, randomly sampling from the aforementioned annotated instances. We evaluate our system on the development split of FEVER (Thorne et al., 2018), consisting of $19,998$ claims, using retrieved evidence. We use the document retriever of Aly and Vlachos (2022), and the sentence reranker of Stammbach (2021) to select the top $k = 5$ evidence sentences $E$. To assess the robustness of QA-NatVer, we also evaluate the systems on Symmetric FEVER (Schuster et al., 2019), a binary classification dataset (Supports, Refutes) consisting of 712 instances, which is built to expose models that learn artefacts and erroneous biases from FEVER.

**Baselines** As a state-of-the-art faithful inference system, ProoFVer (Krishna et al., 2022) is the main baseline we compare against, which is based on GENRE (Cao et al., 2021), an end-to-end entity linking model, fine-tuned on BART (Lewis et al., 2020). We evaluate a version of ProoFVer that is trained on the same data as QA-NatVer to compare both system's data efficiency. We refer to the version of ProoFVer trained on over 140, 000 FEVER instances using additional knowledge sources as outlined in Sec. 1) as ProoFVer-full. Moreover, we evaluate in our few-shot setting LOREN (Chen et al., 2022a), which decomposes claims into phrases, and predicts their veracity using a neural network regularized on the latently encoded phrases' veracity by simple logical rules. Similarly to ProofVer, LOREN was trained on the entirety of FEVER.

We further consider few-shot baselines that do not guarantee faithfulness or provide the same level of interpretability. These include a state-of-the-art few-shot learning method, T-Few (Liu et al., 2022), which uses two additional loss terms to improve few-shot fine-tuning. While Liu et al. (2022) is based on T0 (Sanh et al., 2022), we instead use BART0 (Lin et al., 2022), a BART model instruction-finetuned on the multi-task data mixture described in Sanh et al. (2022), to keep the baselines comparable. They observe comparable performance between BART0 and T0, which we confirmed in preliminary experiments. Finally, we also evaluate a finetuned DeBERTa$_{LARGE}$ model (He et al., 2021), and a DeBERTa model additionally fine-tuned on SNLI (Bowman et al., 2015) and MultiNLI (Williams et al., 2018), both being common and very competitive claim verification baselines (Stammbach, 2021; DeHaven and Scott, 2023).

**Experimental Setup** We are sampling $K$ training samples and do not consider a validation set for hyperparameter-tuning, following the real-world few-shot learning setting of Alex et al. (2021). We use Awesomealign (Dou and Neubig, 2021) as the word-level contextualized alignment system. To stay comparable with our ProoFVer and T-Few baselines, we also use BART0 as our instruction-tuned language model. Notably, natural language inference is not a task BART0 has seen during instruction fine-tuning. To take advantage of a more powerful QA system, we also evaluate QA-NatVer using Flan-T5 (Chung et al., 2022), a state-of-the-

art instruction-tuned language model, which has explicitly seen natural language inference amongst many more tasks. Results are averaged over five runs with standard deviation indicated unless otherwise noted.

|  | Accuracy | Macro-Avg. $F_1$ |
|---|---|---|
| DeBERTa | $39.1 \pm 2.5$ | $36.7 \pm 2.3$ |
| DEBERTa-NLI | $59.7 \pm 1.9$ | $59.4 \pm 1.6$ |
| T-Few | $59.4 \pm 3.2$ | $58.8 \pm 3.1$ |
| LOREN | $35.7 \pm 1.8$ | $27.9 \pm 1.7$ |
| ProoFVer | $36.2 \pm 1.3$ | $32.5 \pm 1.3$ |
| QA-NatVer | $64.0 \pm 0.9$ | $63.1 \pm 1.5$ |
| + Flan-T5 | $\mathbf{70.3 \pm 2.1}$ | $\mathbf{70.0 \pm 1.4}$ |

Table 2: Results on FEVER with 32 claim annotations for training.

**Main Results** Results are shown in Figure 2. QA-NatVer achieves an accuracy of $64.0$ and a macro-averaged $F_1$ of $63.1$, outperforming T-Few by $4.6$ and $4.3$ absolute points respectively. More importantly, our system substantially outperforms ProoFVer which performs only marginally better than random when trained on the same amount of data. We improve accuracy by $27.8$ points, highlighting the limitation of the ProoFVer formulation in a few-shot scenario, and this observation also holds for LOREN. Despite QA-NatVer's BART0 not having seen any natural language inference task in its instruction-tuning, it outperforms the DEBERTav3-NLI model. When training our QA-NatVer model with the Flan-T5 model instead, our system accuracy improves to $70.3 \pm 2.1$ with an $F_1$ of $70.0 \pm 1.4$ while being trained only on 32 claim annotations. For comparison, ProoFVer-full, achieves an accuracy of $78.5$, highlighting the data efficiency of QA-NatVer.

|  | Accuracy | Macro-Avg. $F_1$ |
|---|---|---|
| DeBERTa | $52.3 \pm 4.0$ | $47.5 \pm 8.0$ |
| DEBERTa-NLI | $83.1 \pm 1.6$ | $81.4 \pm 0.9$ |
| T-Few | $73.7 \pm 3.8$ | $73.0 \pm 4.7$ |
| ProoFVer | $53.4 \pm 2.7$ | $52.6 \pm 2.7$ |
| QA-NatVer | $80.9 \pm 1.2$ | $80.9 \pm 1.2$ |
| + Flan-T5 | $\mathbf{85.8 \pm 0.9}$ | $\mathbf{85.8 \pm 1.0}$ |

Table 3: Results on Symmetric-FEVER.

**Robustness** As shown in Table 3, QA-NatVer performs competitively against all baselines when run without adjustment on Symmetric FEVER. QA-NatVer outperforms ProoFVer by $28.7$ accuracy

points and T-Few by 7.2 points. Our system is also competitive with models trained on the entirety of FEVER, including ProoFVer-full, performing 1.2 accuracy points worse than ProoFVer-full (82.1). Training DeBERTa on NLI datasets improves robustness substantially, performing even better than ProoFVer-full. Using Flan-T5 as our instruction-tuned model instead, QA-NatVer surpasses all other models, improving scores by about 4.9 accuracy points to reach an accuracy of $85.8 \pm 0.9$.

**Varying sample sizes.** Table 4 compares our approach against the baselines when trained with varying amounts of data in a few-shot setting, ranging between 16, 32, and 64 samples. QA-NatVer consistently outperforms our baselines across sample sizes. Notably, while DeBERTav3 sees improvements with 64 samples, ProoFVer's improvements are marginal, indicating a larger need for data. The variability decreases across all models with increasing sample size, with QA-NatVer having the lowest standard deviation, indicating its training robustness. QA-NatVer with Flan-T5 achieves a sore of $71.6 \pm 1.7$ when trained with 64 samples. The question-answering formulation might also be beneficial to obtaining large-scale proof annotations (cf. Section 2 cheaply, reducing the workload for the annotators. Investigating this option is left to future work.

| Num. Samples | 16 | 32 | 64 |
|---|---|---|---|
| DeBERTav3 | $36.5 \pm 2.4$ | $39.1 \pm 2.5$ | $47.4 \pm 5.6$ |
| DeBERTav3-NLI | $55.2 \pm 3.0$ | $59.7 \pm 1.9$ | $63.4 \pm 1.0$ |
| T-Few | $53.4 \pm 5.9$ | $59.4 \pm 3.2$ | $61.6 \pm 1.1$ |
| ProoFVer | $33.7 \pm 0.5$ | $36.2 \pm 1.3$ | $37.7 \pm 1.3$ |
| QA-NatVer | $\mathbf{60.2 \pm 2.6}$ | $\mathbf{64.0 \pm 0.9}$ | $\mathbf{65.5 \pm 0.4}$ |

Table 4: Results on FEVER with 16, 32, and 64 samples.

**Claim length.** Real-world claims are typically longer than the example claim shown in Figure 1. MultiFC (Augenstein et al., 2019), a large-scale dataset of real-world claims, has an average claim length of 16.7 tokens compared to FEVER with 9.4. We subsequently measure the performance of QA-NatVer as a function of a claim's minimum length, as shown in Figure 3. QA-NatVer shows only a very small performance decline for claims of up to a minimum of 18 tokens, indicating its robustness towards longer claims, correctly predicting the veracity of claims such as "*The abandonment of populated areas due to rising sea levels is caused by global warming*".

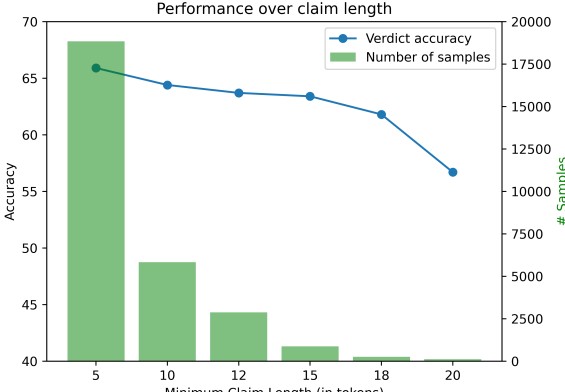

Figure 3: QA-NatVer on claims of varying length.

**Ablation** We perform three different ablation studies reported in Table 5. First, we examine the performance of QA-NatVer without multi-granular chunking. We observe a 7.7 average score drop in accuracy, demonstrating that considering evidence spans at different levels of granularity improves performance. Second, we ablate our proof selection method by omitting the veracity score $s_v$, observing an accuracy drop by 3.7 points.

Finally, we compare our question-answering approach for NatOp assignments to a model that is prompted to predict NatOPs for claim-evidence spans as a multi-class problem without multi-granular chunking. We use ChatGPT (OpenAI, 2022) and Llama-2 (Touvron et al., 2023) as state-of-the-art few-shot language models and prompt them with in-context examples from our annotated proofs. The other parts of QA-NatVer are kept identical. We observe that the non-QA approach leads to predicting more independence operators (#), resulting in an 11.9 points drop in accuracy with ChatGPT, and a 16.6 drop with Llama2-13B. For details see Appendix D.

| Setup | Acc | $F_1$ |
|---|---|---|
| QA-NatVer | $64.0 \pm 0.9$ | $63.1 \pm 1.5$ |
| w/o multi-granular chunking | $56.3 \pm 0.6$ | $55.3 \pm 0.7$ |
| w/o verdict score $s_v$ | $60.3 \pm 1.4$ | $57.1 \pm 3.2$ |
| w/o QA (w/ ChatGPT) | $52.1$ | $51.9$ |
| w/o QA (w/ Llama2-13B) | $47.4$ | $42.4$ |

Table 5: Ablation study of QA-NatVer on FEVER.

## 4.2 Application to Lower-Resource Language

**Data** To assess QA-NatVer's performance for languages with fewer resources than English, we evaluate it without further training (apart from the 32 FEVER annotated English claims) on DanFEVER

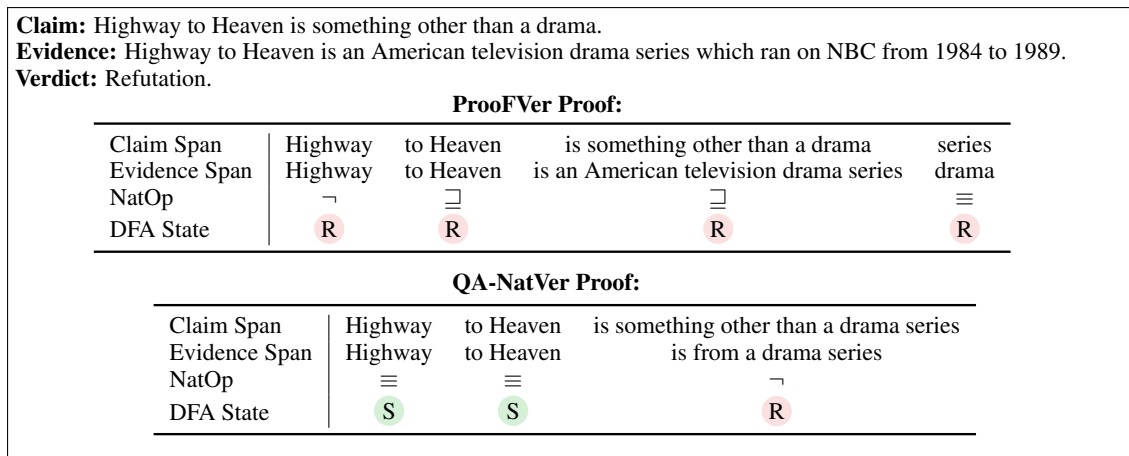

Figure 4: A FEVER example where ProoFVer and QA-NatVer reach the correct verdict (refutation). QA-NatVer produces more plausible proofs with fewer erroneous NatOp assignments than ProoFVer.

(Nørregaard and Derczynski, 2021), a three-way claim verification task for Danish, consisting of a total of 6,407 instances. To eliminate additional variability from a multilingual retrieval system, we use the gold evidence for evaluation, except for NEI-labeled instances for which we retrieve evidence via BM25.

|  | Accuracy | Macro-Avg. $F_1$ |
|---|---|---|
| XLM-RoBERTa | $41.9 \pm 3.0$ | $31.7 \pm 3.3$ |
| XLM-ROBERTa-XNLI | $54.3 \pm 1.3$ | $52.2 \pm 0.7$ |
| T-Few | $59.2 \pm 4.8$ | $53.8 \pm 12.7$ |
| ProoFVer | $47.9 \pm 3.2$ | $24.6 \pm 2.9$ |
| ProoFVer-full w/ MT | $56.9$ | $52.1$ |
| QA-NatVer | $\mathbf{61.0 \pm 3.5}$ | $\mathbf{56.5 \pm 4.9}$ |

Table 6: Results on DanFEVER, using 32 FEVER claims using a multilingual language model.

**Baselines & Experimental Setup:** We use our baselines with multilingual backbones, namely T-Few, and ProoFVer with an mT0 backbone (Muennighoff et al., 2022), as well as a finetuned XLM-RoBERTa (Conneau et al., 2020) model. We additionally consider ProoFVer-full by translating the claim and evidence from Danish into English, using the translation system by Tiedemann and Thottingal (2020). QA-NatVer also uses mT0 (Muennighoff et al., 2022), a multilingual T5-based model, instruction-tuned on multilingual tasks. Similarly to BART0, mT0 has not seen natural language inference in its fine-tuning. We use the chunking system of Pauli et al. (2021) and keep all other components unchanged. The language of the natural logic questions and answers remains English for all experiments.

**Results** Results on DanFEVER are shown in Table 6. Our system achieves accuracy and $F_1$ of 61.0 and 56.5, outperforming all other baselines by 1.8 and 2.7 points, respectively. The ProoFVer baseline trained on the same data as our model achieves a score of 47.9. Notably, in this setting our approach even outperforms ProoFVer-full, where the claims and evidence being translated from Danish into English. Running ProoFVer-full in this setting is computationally expensive due to the translation required and still has worse accuracy than QA-NatVer. The variability in this language transfer setup is higher than for FEVER, particularly for T-Few, but remains low for QA-NatVer.

### 4.3 Correctness of Natural Logic Operators

Assessing the quality of generated proofs exclusively by the verdict they result in, ignores that an incorrect proof might lead to the correct verdict. For instance in Figure 4, ProoFVer fails to assign equivalence ($\equiv$) even to identical spans, such as *Highway* and *Highway*, yet it still produces the correct veracity label. To intrinsically evaluate the quality of proofs, human subjects (not the authors of this paper) annotated a total of 114 NatOp assignments from 20 claims and their associated proof from both ProoFVer and QA-NatVer for their correctness. Each NatOp assignment was annotated by 3 annotators, resulting in 342 data points. The claims are selected via stratified sampling, ensuring that each class is equally represented. We further ensure that both models predict the same verdict. All three subjects assigned the same correctness label to a NatOp in 84.8% of cases, thus indicating high inter-annotator agreement. QA-NatVer's

NatOp assignments are correct in $87.8\%$ of the cases, while ProoFVer is only correct in $63.4\%$, indicating that the quality of NatOp assignments by QA-NatVer is superior to those by ProoFVer.

Considering the very high label accuracy of ProoFVer (outperforming QA-NatVer by almost 10 accuracy points), these results are surprising. We hypothesise that ProoFVer might have learned "shortcuts" to arrive at the correct verdict in its proof due to the noisy signals in the weakly supervised proof dataset it has been trained on, due to the dataset-specific heuristics that have been applied to construct a dataset of sufficient size to train it on. To validate this, we inspect mutations where the claim and evidence spans are identical. These are trivial cases where the model is expected to predict the equivalence operator. However, ProoFVer produces a wrong NatOp for about $16.3\%$ of cases, mostly the independence operator ($13\%$), but our system always predicts equivalence (see App. C).

## 4.4 Plausibility

To assess the plausibility of the natural logic proofs predicted by QA-NatVer, we run a forward prediction experiment (Doshi-Velez and Kim, 2017). Human subjects are asked to predict the veracity label *solely* from the justification (i.e. proof) generated by the model and to specify on a five-level Likert scale, ranging from *very plausible* to *not plausible*, how plausible the justification appears to them. Since we are evaluating proofs as an explanatory mechanism to humans, we ensured that no subject was familiar with the deterministic nature of natural logic inference. To enable non-experts to make use of the proof, we replaced the NatOps with English phrases, similar to (Krishna et al., 2022) (see Appendix C).

The evaluation consists of 120 annotations from 6 subjects. The same 20 claims used in the human evaluation of correctness are paired with a ProoFVer or QA-NatVer proof explanation and are annotated by three subjects. No subject annotates the same claim for both models, as otherwise a subject might be influenced by the explanation it has seen before for the same claim. Using the QA-NatVer proofs, subjects correctly predict the model's decision in $90\%$ of cases, compared to ProoFVer's $76.9\%$. All three subjects selected the same verdict in $70\%$ and $91.7\%$ of the cases, for ProoFVer and Qa-NatVer, respectively, with an inter-annotator agreement of $0.60$ and $0.87$ Fleiss

$\kappa$ (Fleiss, 1971). Regarding the plausibility assessments, the subjects rate QA-NatVer proofs an average score of $4.61$, while ProoFVer is rated $4.16$ out of $5$ points.

## 4.5 Efficiency

QA-NatVer remains computationally efficient since the typical bottleneck of transformer models, the input and output length, remain short at every stage of the pipeline. Concretely, the alignment module encodes each evidence sentence with the claim independently. The QA model uses as input a question with a single embedded claim span and its evidence with the output being either Yes/No or a short phrase. The average input length to the QA model on FEVER is 20 tokens while its output is in most cases a single token. This is substantially cheaper computationally than cross-encoding the claim and all evidence sentences and autoregressively generating the proof at once, as done by ProoFVer, with 195.2 input and 31.1 output tokens on average. The entire runtime of the QA module can be described as $\mathcal{O}(l \cdot n_{span}^2 + n_{all}^2)$, with $l$ being the number of spans, $n_{span}$ being the input length for the aligned claim-evidence spans (for the NatOp probability score) and $n_{all}$ being the length of the claim and its evidence sentences (for the NatOp verdict score). We measure the wall-clock time (in minutes) with the BART-large backbone, using the same hardware configuration as described in Appendix B. DeBERTa, T-Few, ProoFVer, LOREN, and QA-NatVer train in 5.3, 22.3, 21.4, 27.5, and 36.4 minutes, and inference on the FEVER development set of 19998 instances runs in 20.6, 7.3, 185.2, 116.5, and 89.1 minutes, respectively.

## 5 Conclusion

This paper presented QA-NatVer, a natural logic inference system for few-shot fact verification that frames natural logic operator prediction as question-answering. We show that our approach outperforms all baselines while remaining faithful. Human evaluation shows that QA-NatVer produces more plausible proofs with fewer erroneous natural logic operators than the state-of-the-art natural logic inference system, while being trained on less than a thousandth of the data, highlighting QA-NatVer's generalization ability. Future work looks at extending the capability of natural logic inference systems to more complex types of reasoning, including arithmetic computations.

## Limitations

While natural logic provides strong explainability by operating directly on natural language, it is less expressive than alternative meaning representations that require semantic parses such as lambda calculus (Zettlemoyer and Collins, 2005). For instance, temporal expressions and numerical reasoning are beyond the expressive power of natural logic (Krishna et al., 2022) but are frequently required when semi-structured information is available (Aly et al., 2021). Moreover, cases of ambiguity like cherry-picking, are difficult to process with natural logic. Addressing the limits of natural logic inference systems is out-of-scope for this paper. Similar to ProoFVer, the proof we constructed is intended to be executed in the DFA from left to right, however, natural logic-based inference is not constrained to such execution. Furthermore, all benchmarks explored in the paper use Wikipedia as the knowledge base which is homogeneous compared to heterogeneous sources professional fact-checkers use (e.g., news articles, scientific documents, images, videos)

## Ethics Statement

Our system improves the explainability of claim verification models and empowers actors to make more informed decisions about whether to trust models and their judgements, yet actors must remain critical when evaluating the output of automated claim verification systems and not confuse explainability with correctness. We emphasize that we do not make any judgements about the truth of a statement in the real world, but only consider Wikipedia as the source for evidence to be used. Wikipedia is a great collaborative resource, yet it has mistakes and noise of its own similar to any encyclopedia or knowledge source. Thus we discourage users using our verification system to make absolute statements about the claims being verified, i.e. avoid using it to develop truth-tellers.

## Acknowledgements

This work was supported by the Engineering and Physical Sciences Research Council Doctoral Training Partnership (EPSRC). Andreas Vlachos is supported by the ERC grant AVeriTeC (GA 865958) and the EU H2020 grant MONITIO (GA 965576). Further, the authors would like to thank the subjects who volunteered to be part of the human evaluation, namely Mubashara Akhtar, Julius Cheng, Sana Kidwai, Nedjma Ousidhoum, Andre Schurat, and Ieva Raminta Staliunaite. We finally thank the anonymous reviewers for their time and effort giving us feedback on our paper.

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

## A  QA-NatVer

**Question-Answering System**  BART0 is a pretrained seq2seq model which has been instruction-finetuned to autoregressively generate the prediction for a given input prompt. In the case of boolean questions where only a single token is predicted, the generation can be simplified to Equation 1. Otherwise, such as for equation 3, where the answer coice $y \in \{\text{Supports}, \text{Reftues}, \text{Not enough info}\}$, is longer than a single token, the probability assigned QA-NatVer can be described via an autoregressive formulation as:

$$\text{QA}(y \mid c, E, T_v) =$$
$$\frac{1}{|T|}\frac{1}{|M|}\sum_m \sum_{t \in T_v} \log p_\theta(y \mid c, E, t, y_{<m}), \quad (4)$$

with $M$ being the length of the textual representation of $y$.

The input is structured so that the claim is followed by the evidence sentences, each separated by end-of-sentence tokens: c  $e_0$,  ...  $e_k$. Each evidence sentence in $E_i$ is preceded by the corresponding document title in square brackets, e.g. "*[James McBrayer] Jack McBrayer (born May 27... [Tom Bergeron] Tom Bergeron (born May 6, 1955) ...*".

**Training**  We follow Liu et al. (2022) when finetuning our model on multiple prompts, randomly selecting a prompt $t \in T_o$. We fine-tune all weights of the model (both for BART0 and Flan-T5). We experimented with sampling negative training instances for QA-NatVer that follow the positive to negative sample distribution as present during inference (i.e. 1:4 positive to negative question-answer pairs), yet the training process frequently diverged due to the high label imbalance. The question-answering module is trained jointly on veracity question templates and natural logic operator templates.

## B  Implementation Details

All models are implemented using PyTorch (Paszke et al., 2019), making use of PyTorch Lightning (Falcon, 2019). All experiments using BART0 as the instruction-finetuned QA model were run on a machine with a *single* Quadro RTX 8000 with 49GB memory and 64GB RAM memory. To fine-tune the Flan-T5 model (Flan-T5-xl) with its 3 billion parameters, we use a single Ampere A100 GPU with 80GB memory. We have also run some early tests with T0 (Sanh et al., 2022) but noticed that the performance of BART0 is comparable, confirming observations made in Lin et al. (2022). All models are implemented using Huggingface (Wolf et al., 2020).

**QA-NatVer**  The language model we used with AwesomeAlign is the multilingual BERT model (Devlin et al., 2019). To fine-tune our word alignment system, we use

```
 1   Determine an entailment relation between two expressions.
 2
 3   Use one of the following entailment relations: Equivalence, Negation, Alternation, Forward–Entailment, Reverse–
     Entailment, Independence.
 4
 5   Equivalence means that two expressions have the same meaning. This includes synonyms and paraphrases.
 6   Examples of Equivalence:
 7   {N examples of Equivalence}
 8
 9   Negation means that one expression negates the other and that both cannot be true at the same time.
10   Examples of Negation:
11   {N examples of Negation}
12
13   Alternation means that expressions are mutually exclusive.
14   Examples of Alternation:
15   {N examples of Alternation}
16
17   Forward–Entailment means that the first expression entails the second expression.
18   Examples of Forward–Entailment:
19   {N examples of Forward–Entailment}
20
21   Reverse–Entailment means that the second expression entails the first expression.
22   Examples of Reverse–Entailment:
23   {N examples of Reverse–Entailment}
24
25   Independence means that there is no informative relation between expressions.
26   Examples of Independence:
27   {N examples of Independence}
```

Listing 1: The instructive part of the prompt used in ChatGPT and Llama2 ablation experiments. For each NatOp, we provided a short explanation and 5 examples (colored in blue) formatted as *{evidence,claim,natop}* triples.

the AwesomeAlign repository[5]. the sentence transformer we use for alignment is `sentence-transformers/all-mpnet-base-v2`. In addition to the similarity score from the sentence transformer to select the most appropriate evidence span $e_i$ from all options $e_{ij}$, we take into account the evidence retriever's ranking of the evidence sentences, down-weighting the similarity score by a factor that scales negatively with the rank, i.e. $(j \cdot 0.8)$.

**Baselines**  We evaluate the T-Few baseline using the provided repository[6], which also provided the basis for the QA-NatVer code implementation. Liu et al. (2022) also propose a parameter-efficient fine-tuning method (named $(IA)^3$) in addition to the loss functions, however, we observe much more stable training loss and better results in preliminary experiments when tuning the entire model, particularly for BART0. This observation of degrading performance on BART0 with $(IA)^3$ is also observed in Aly et al. (2023). Therefore, for all experiments, and all models (baselines + QA-NatVer) we trained all model weights. Krishna et al. (2022) kindly

provided us access to their ProofVER model. For our ablation experiment with ChatGPT, We use OpenAI's API (Brockman et al., 2020) to query *gpt-3.5-turbo-0613*, and ask ChatGPT to assign NatOPs to batches of claim-evidence pairs (at most 25 spans per query). For ablation experiments with Llama2, we ran $13B$ parameter models locally. We used the GPTQ (Frantar et al., 2022) version of these models with 4-bit quantization to lower the computational requirements and speed up the inference.

**Hyperparameters**  Since no development data was used to tune hyperparameters, we set them to the default values described in Liu et al. (2022). We only reduced the learning rate for QA-NatVer as we noticed that the training loss was unstable. Specifically, we set the hyperparameters to `learning_rate: 1e-5`, `batch_size: 8`, `grad_accum_factor: 4`, `training_steps: 2000`, use the AdamW optimizer (Loshchilov and Hutter, 2019), and a linear decay with warmup scheduler. The same hyperparameters are used with Flan-T5.

---

[5]https://github.com/neulab/awesome-align
[6]https://github.com/r-three/t-few

## C  Human Evaluation

All subjects in the human evaluation are graduate/-postgraduate students in either computer science or linguistics. 3 subjects are male, 3 female. None of the subjects had prior knowledge of natural logic inference. Table 7 shows the textual description used for the NatOps in the human evaluation and Table 8 shows the predicted NatOps by both ProoFVer and our system for aligned claim-evidence spans that are identical.

| NatOP | Textual Description |
|:---:|:---:|
| ≡ | Equivalent Spans |
| ⊑ | Claim span follows from the evidence span |
| ¬ | Claim span is negated by the evidence |
| ⇃↾ | Evidence span contradicts the claim span |
| ⊒ | Incomplete Evidence |
| # | Unrelated claim span and evidence span |

Table 7: NatOPs and their corresponding textual description as shown to the human annotators, following (Krishna et al., 2022).

| | ProoFVer | Ours |
|:---:|:---:|:---:|
| # Eq. Spans | 9832 | 17774 |
| Eq. Natop | 8233 (83.7%) | 17774 (100%) |
| Fwd. Entail Natop | 84 (1%) | 0 (0%) |
| Rev. Entail Natop | 193 (2%) | 0 (0%) |
| Neg Natop | 0 (0%) | 0 (0%) |
| Alt Natop | 48 (0.5%) | 0 (0%) |
| Indep Natop | 1274 (13%) | 0 (0%) |

Table 8: Predicted NatOps for claim and evidence spans that are identical. Despite the triviality, ProoFVer does assign the wrong operator in about 16.3% of cases. In contrast, our system does consistently predict equality in every instance.

## D  Prompting

Listing 1 shows a template for the instructive part of our prompts with ChatGPT and Llama2. For ChatGPT, we used OpenAI's API (Brockman et al., 2020) to query *gpt-3.5-turbo-0613*, and asked ChatGPT to assign NatOPs to claim-evidence pairs. We used batches of at most 25 spans per query to lower the running costs. For Llama2 experiments, we ran the models locally and asked the models to assign one NatOP per query.