# OpenReview forum: "QA-NatVer: Question Answering for Natural Logic-based Fact Verification"
_EMNLP/2023/Conference — EMNLP 2023 Main_

### Official Review · Reviewer_5v7q · 2023-07-25

**Soundness:** 4

**Excitement:**

3: Ambivalent: It has merits (e.g., it reports state-of-the-art results, the idea is nice), but there are key weaknesses (e.g., it describes incremental work), and it can significantly benefit from another round of revision. However, I won't object to accepting it if my co-reviewers champion it.

**Missing References:**

[1] Logic-Regularized Reasoning for Interpretable Fact Verification. AAAI 2022.

**Paper Topic And Main Contributions:**

This paper proposes QA-NatVer, a natural logic inference system for fact verification that frames natural logic operators in a QA setting.
The main contribution of the paper is a dynamic programming approach to proof selection that uses a multi-granular merging mechanism to construct proofs from non-overlapping claim chunks.
The paper also introduces a scoring function that combines NatOp probability scores and NatOp verdict scores to decode the optimal assignment.
Overall, it provides a novel approach for fact verification that can improve the accuracy and explanation of the process.

**Questions For The Authors:**

1. The proof selection part isn't very clear, is this similar to the CYK algorithm?
The authors should give state transition equations or figure to clarify it.

**Reasons To Accept:**

1. The paper presents a novel approach to fact verification that combines natural logic with question answering, which is a big contribution to the field.

2. The proposed system outperforms existing baselines and state-of-the-art systems in terms of accuracy in a few-shot setting on FEVER.

**Reasons To Reject:**

1. There are too few proof annotations on the test set (only 32), which weakens the solidity of the work.

2. Multiple baselines use different pre-trained models, and this paper also uses external tools, the experimental comparison is not very fair.


**Reproducibility:**

3: Could reproduce the results with some difficulty. The settings of parameters are underspecified or subjectively determined; the training/evaluation data are not widely available.

**Reviewer Confidence:**

4: Quite sure. I tried to check the important points carefully. It's unlikely, though conceivable, that I missed something that should affect my ratings.

---

> ### Author Rebuttal · Authors · 2023-08-28
>
> We would like to thank the reviewer for the time and comments on our paper! We address the reviewer's questions and the modifications we make to our paper based on the reviewer's suggestions below.
>
> **There are too few proof annotations on the test set (only 32), which weakens the solidity of the work.**
>
> We believe the reviewer meant the training set (we always evaluate on the full test set of 19998 instances). In Table 4, we also show performance when training our method and the baselines on 64 instances. We observe a clear and consistent performance improvement, from 16, 32, to 64 samples. 64 samples is often considered an upper limit for few-shot learning (see for instance Alex et al. 2021 [1])
>
> **Multiple baselines use different pre-trained models, and this paper also uses external tools, the experimental comparison is not very fair.**
>
> To clarify: we generally used the same pre-trained model, but deviated from it when it was *in favour* of the baselines:
>
> * T-Few, as the most powerful autoregressive approach, uses the same pre-trained model as QA-NatVer (BART0).
>
> * DeBERTav3+NLI, as the strongest three-way classifier, performs much better than BART0 when used directly as a classifier (41.6 versus 59.7 accuracy).
>
> * ProofVer uses GENRE (BART fine-tuned on entity-linking [2]) as it performed better than BART0 (only 34.3 accuracy).
>
> Assuming the reviewer is referring to the chunking and alignment system as the external tool, we would like to emphasize that these are commonly used models for these tasks and are typically incorporated into more complex pipelines like ours. Other baselines make use of tools as well and even additional data, such as ProoFVer with its entity-linking component (GENRE) or DeBERTav3 with the MNLI training data.
>
> **Question 1: The proof selection part isn't very clear, is this similar to the CYK algorithm?**
>
> We will clarify the description of our proof selection in our paper. The selection is not similar to CYK, instead, the dynamic program is only used to construct all possible proofs efficiently. Each proof is then scored individually via a NatOp probability score (Eq. 2) and a NatOp verdict score (Eq. 3). Note that all proofs are constructed from 4 x l − 6 aligned claim-evidence spans as defined in section 3.1 (with l being the number of chunks the claim is initially decomposed into). The same span is used across multiple proofs but its NatOp probability score is only computed once, for efficiency reasons. For a more detailed discussion on QA-NatVer’s efficiency see our response to reviewer 1.
>
> **Missing Reference: LOREN**
>
> We thank the reviewer for the additional reference and will incorporate it into our related work, see our response to reviewer 1 (in the section "Missing reference: LOREN") for comparison against it.
>
> [1] https://datasets-benchmarks-proceedings.neurips.cc/paper/2021/file/ca46c1b9512a7a8315fa3c5a946e8265-Paper-round2.pdf

---

### Official Review · Reviewer_Esj4 · 2023-08-02

**Soundness:** 4

**Excitement:**

4: Strong: This paper deepens the understanding of some phenomenon or lowers the barriers to an existing research direction.

**Paper Topic And Main Contributions:**

The authors propose a system for constructing a proof of a claim from a set of evidence statements using Natural Logic. However, rather than use traditional natural logic operators (where the applicability of a particular operator is determined by a model trained on a large amount of data), the authors cast the operators into questions. This allows their approach to use strong pretrained language models to assess whether a particular operator applies when constructing a proof. Consequently, their approach requires less labeled data and exhibits stronger performance

**Reasons To Accept:**

The paper is well-written and straightforward to follow.

The focus on designing language model-based systems that produce explanations that can be trusted is appropriate given the current emphasis on pure LLM-based systems.

The emphasis on low-resource languages is good to see

The paper thoroughly tests their method, showing performance across 1) multiple datasets, 2) different languages, and 3) has a human evaluation

**Reasons To Reject:**

The main limitation in my opinion would be that the range of situations where this method is applicable would be limited to situations that can be handled by natural logic. I do not consider this a significant issue, however.

**Reproducibility:**

3: Could reproduce the results with some difficulty. The settings of parameters are underspecified or subjectively determined; the training/evaluation data are not widely available.

**Reviewer Confidence:**

3: Pretty sure, but there's a chance I missed something. Although I have a good feel for this area in general, I did not carefully check the paper's details, e.g., the math, experimental design, or novelty.

---

> ### Author Rebuttal · Authors · 2023-08-28
>
> We thank the reviewer for the time and comments on our paper! Indeed, our approach inherits the limitations of Natural Logic (see Krishna et al. 2022 [1] for a discussion), and we will mention this in the camera ready if accepted.
>
> [1] https://aclanthology.org/2022.tacl-1.59/

---

### Official Review · Reviewer_5F8N · 2023-08-02

**Typos Grammar Style And Presentation Improvements:** None
**Soundness:** 3

**Excitement:**

3: Ambivalent: It has merits (e.g., it reports state-of-the-art results, the idea is nice), but there are key weaknesses (e.g., it describes incremental work), and it can significantly benefit from another round of revision. However, I won't object to accepting it if my co-reviewers champion it.

**Missing References:**

1. The claim in Line 151-153 that "these works do not consider aggregating nor constructing proofs from the answers to the questions" is not very accurate. In [LOREN: Logic-Regularized Reasoning for Interpretable Fact Verification](https://ojs.aaai.org/index.php/AAAI/article/view/21291) (Chen et al., in AAAI 2022), they adopted a very similar idea to this work: decompose claims (chunking) and acquire veracity information from evidence with QA, construct proofs with them, and aggregate them with logic regularization. I'd suggest the authors to include more discussions with this work and its ideas.

**Paper Topic And Main Contributions:**

This paper primarily investigates how to use QA to predict natural logic operators to improve fact verification systems. The main contributions include:
1. Proposing a novel method of using QA to predict natural logic operators, which avoids the need for extensive annotated training data while still relying on deterministic reasoning systems.
2. Introducing the QA-NatVer system, a natural logic reasoning system that constructs proofs by transforming natural logic into a QA framework.
3. Outperforming all baselines in a low-resource setting on FEVER, including a state-of-the-art pre-trained seq2seq natural logic system and a state-of-the-art prompt-based classifier. The system demonstrates its robustness and portability on adversarial datasets and Danish verification datasets.

**Questions For The Authors:**

QA. How does this method perform on complex claims?

QB. Have you run any experiments where BART0 and Flan-T5 are directly fine-tuned for fact-checking tasks? I expect them to make a strong baseline, especially for Flan-T5 models.

QC. What's the baseline performance of LLMs (like InstructGPT or ChatGPT) in this task?

QD. Can you design a chain-of-thought prompting for LLMs that resembles the NatOps and Proofs used in the proposed method? You can also add several annotated in-context examples for this purpose. I'm curious whether such decomposition and reasoning method could be useful for LLMs, because the reasoning process seems to be the major contribution of this work.

QE. How is the efficiency of this method?

QF. How does the size of backbone model (i.e. scaling law) influence the effectiveness of this method?

QG. How is the expressiveness of natural logic, and what is the limitation of it? Can LLMs reason properly with natural logic?

**Reasons To Accept:**

The strengths of this paper include:
1. This paper proposes a reasonable method of using QA to predict natural logic operators, contributing to the explainability of neural fact verification systems.
2. This paper describes in detail about the design and implementation of the QA-NatVer system, including the process of proof construction, which is very helpful for understanding and reproducing the system.
3. The system performs well on multiple datasets, and enjoys an additional edge over previous method on the correctness and plausibility of proofs.

**Reasons To Reject:**

1. Lack of discussions on LLMs. I would really love to see how this idea could be integrated into current large language models (LLMs), like InstructGPT and ChatGPT, which have strong few-shot in-context learning and chain-of-thought reasoning abilities. Conveniently, the major ideas and techniques in this paper can be easily adapted to the LLM CoT prompting setting (few-shot, reasoning chains, etc.). In a time when the foundations of most systems are shifted toward LLMs, I think it's important to add them into discussions. I would really love to see if smaller models could beat LLMs using some good designs.
2. The proposed method seems to only work on simple claims in FEVER. However, it is unknown if naturally-occurring claims with multiple sentences and complex structures can be correctly processed by this method, especially for the chunking & alignment module.
3. The pipeline of this method seems to be a little lengthy, which raises concerns for its practical value. Adding efficiency analysis would be helpful to clarify this.
4. I somehow hold concerns about natural logic on real-world scenarios, for example, claims with first-order logic like "not all US senators are hispanic".

**Reproducibility:**

4: Could mostly reproduce the results, but there may be some variation because of sample variance or minor variations in their interpretation of the protocol or method.

**Reviewer Confidence:**

4: Quite sure. I tried to check the important points carefully. It's unlikely, though conceivable, that I missed something that should affect my ratings.

---

> ### Author Rebuttal · Authors · 2023-08-28
>
> We thank the reviewer for the time and comments on our paper! We address the reviewer's questions and the modifications we make to our paper based on the reviewer's suggestions below.
>
> **Lack of discussions on LLMs (Question C, Question D, Question F)**
>
> We agree with the reviewer that LLMs are worth taking into account and that their integration into a natural language inference system is of interest. In fact, our paper already contains some experiments using LLMs and how their abilities can be integrated into our system:
>
> 1. In our ablation (lines 469- 479) we investigated how our question-answering formulation for the prediction of NatOps compares to using ChatGPT for the same task, the latter being prompted in a multi-class setting with 5 in-context examples for each NatOp and a short description. Our system outperforms ChatGPT as the latter overpredicts independence operators (#), indicating that our design carries merit (the details of our experiments with ChatGPT will be added in the camera ready if accepted). We will conduct further experiments exploring various prompting strategies to predict NatOps as suggested by the reviewer (chain-of-thought prompting, self-correcting prompting, etc.) and also how LLMs can compose predicted NatOps into proofs. These experiments have been difficult to conduct given the time constraint of the rebuttal but we will report additional findings in the paper.
>
> 2. We show that our approach benefits from more powerful backbones, as we see substantial performance improvements when scaling up from BART0 with 400M to FlanT5-xl with 3B parameters (one might already consider the latter an LLM), highlighting that our method is able to take advantage of recent advancements of LLMs. We have run additional experiments on smaller and intermediate-sized models (BART0-base: 60.1 +- 1.9, and FlanT5-large: 67.1 +-1.4), which further validate the observed trend. Moreover, we managed to improve performance on FlanT5-xl from 69.1 in the last version of the paper to 70.3, with only 32 training samples (64 training samples: 71.6 accuracy points). Our approach with 32/64 samples with the Flant5-xl backbone comes close to state-of-the-art fully supervised models on FEVER, trained on over 140,000 instances. These results highlight how the increased capabilities of LLMs benefit our method. However, due to computational constraints, we have not yet been able to run experiments on the largest Flan-T5 model with 11B parameters. We plan to compute results on this for the camera-ready version if accepted.
>
> **The proposed method seems to only work on simple claims in FEVER. (Question A)**
>
> We agree with the reviewer that some classes of complex claims, for instance, ones that require temporal and numerical reasoning, or cases of ambiguity like cherry-picking, are difficult to process by our method. Yet, these types of claims are difficult to handle by most contemporary verification systems. Further, in most instances, a naturally occurring claim does not span multiple sentences but is instead expressed in a single one, see e.g. MultiFC [1], a large-scale dataset of real-world claims. However, we observe a difference in the claim’s length: MultiFC has an average claim length of 16.7 tokens, versus 9.4 for FEVER. We subsequently measure the performance of QA-NatVer on claims in FEVER that are at least 14 tokens long, which constitute 1286 out of about 20,000 instances. We see only a small performance decline (BART0: 0.62 +- 1.2, FlanT5-xl: 0.67 +- 1.1), indicating that more complicated structures (e.g. “The abandonment of populated areas due to rising sea levels is caused by global warming”) are not problematic per se. We will ensure to add a more detailed analysis of this point into the paper in the additional space if accepted.
>
> **I somehow hold concerns about natural logic on real-world scenarios (Question G)**
>
> We do not attempt to extend the capabilities of natural logic inference systems in our paper.  Instead, we address the severe limitation of faithful inference systems of requiring substantial resources for training. We discuss the limitations of natural logic’s expressiveness in our limitations section. It would be interesting to explore in future work possible ways of mapping more complex semantics (e.g. algorithmic operations) to the set-theoretic semantics of natural logic, but this is beyond the scope of this paper.
>
> **Missing reference: LOREN [...] they adopted a very similar idea to this work [...] constructing proofs with them, and aggregate them via logic regularization.  I'd suggest the authors to include more discussions with this work and its ideas.**
>
> Thank you very much for the additional reference, we will ensure to relate to it in our work!
>
> We conducted experiments for LOREN in a few-shot setting using the experimental setup described in our paper. LOREN requires substantial amounts of training data (it is trained on all of FEVER, 145K instances), thus suffering from similar limitations as other previous work. Without further adjustments (besides upgrading the backbone of the MRC model from BART-base to BART-large to keep the comparison fair), LOREN achieves an accuracy of 33.4 on FEVER. We subsequently attempted to improve the performance by adjusting the hyperparameters to match the ones used in our work (a much higher number of epochs and a smaller learning rate), resulting in a score of 35.7. These results are comparable with ProoFVer in a few-shot setting and confirm the difficulty of building interpretable reasoning systems in low-resource scenarios.
>
> Moreover, the logical rules LOREN defines are similar to the three simple logical rules used in Stacey et al. (2022) [2]. These rules do not constitute a logical proof, as their outcomes are merely aggregated by a neural network component, while in the case of Natural Logic these are aggregated by a finite state automaton. Consider for instance a case of double negation such as for the claim “Paul wants to be dry” and the evidence “Nobody wants to be wet”. While natural logic can easily handle such cases, LOREN’s logical rules are not able to handle these explicitly. Instead, LOREN’s MRC system that generates the “local premises” that the logical rules operate on must be powerful enough to cover for the logical rules’ restrictions (i.e. not generating a local premise that can be refuted since a single refuted local premise results in the refutation of the entire claim,  e.g. not doing a substitution that results in a local premise such as “Paul wants to be wet“). However, shifting the reasoning implicitly to the MRC model results in reduced transparency compared to natural logic, since the double negation is never covered explicitly by the logical rules.
>
> The LOREN comparison will be added to our paper, following further experiments and an analysis of its error sources.
>
> **Have you run any experiments where BART0 and Flan-T5 are directly fine-tuned for fact-checking tasks? (Question B)**
>
> Yes. We initially explored using BART0 as a traditional classifier (similar to DeBERTa, using a linear layer on top), yet observed substantially lower performance than when using the model in the generative setting (i.e. T-Few) (41.6 versus 59.4). For clarification: With T-Few we fine-tune the backbone (BART0) directly for fact-checking, complemented by additional loss functions defined by T-Few. As seen in our paper, the results of T-Few still trail behind QA-NatVer, similar for Flan-T5.
>
>
> **The pipeline of this method seems to be a little lengthy, [...] Adding efficiency analysis would be helpful to clarify this. (Question E)**
>
> We will add a section on our method’s complexity to the paper in the additional space if accepted. Overall, our method remains efficient since the input length to the models at every stage of the pipeline is short, which is typically the computational bottleneck due to the quadratic runtime of transformers with respect to the input length. Concretely, the alignment module encodes each evidence sentence independently with the claim. Similarly, the QA model for the NatOp prediction takes as input a question with a single embedded claim and evidence span. The average input length is 19.95 tokens, substantially shorter than cross-encoding the entire claim and all evidence sentences, with 195.2 tokens on average. The entire runtime of the QA module can be described as  $\mathcal{O}(l * n_{\text{span}}^2 +n_{\text{all}}^2)$, with $n_{\text{span}}$ being the input length for the aligned claim-evidence spans (for the NatOp probability score) and $n_{\text{all}}$ being the length of the entire claim and its evidence sentences (for the NatOp verdict score).
>
> We measure the following wall-clock time (in minutes) training on 32 instances, and running inference on the FEVER development set of 19998 instances, using the same hardware configuration as described in the paper. All models use the BART-large backbone (except DeBERTa):
>
> | Model | Train      | Inference |
> | -----------  | ----------- | ----------- |
> |DeBERTa       |5.3      | 20.6       |
> |T-Few      | 22.3   | 7.31        |
> |ProoFVer     | 21.35   | 185.22        |
> |Loren     | 27.49   | 116.46        |
> |QA-NatVer    | 36.39   | 89.11    |
>
> As shown above, QA-NatVer performs comparably to the baseline methods, taking the longest for training (but still manageable with about half an hour of training time), and less than both LOREN and ProoFVer during inference.
>
> [1] https://aclanthology.org/D19-1475.pdf
>
> [2] https://aclanthology.org/2022.emnlp-main.251.pdf

---

### Meta-Review · Area_Chair_dfCg · 2023-09-27

**Recommendation:** 4

**Metareview:**

The reviewers overall agreed on the novelty of the approach and its good performance on multiple datasets. However, they also raised concerns about the limited applicability of the method to complex claims, the lack of discussions on integrating the approach into large language models, the efficiency of the pipeline, the scarcity of proof annotations on the test set, and the experimental fairness in the comparison with baselines.  The authors have provided a fair amount of discussion to the questions raised by the reviewers.

---

### Decision · Program_Chairs · 2023-10-07

**Decision:**

Accept-Main

**Comment:**

The reviewers overall agreed on the novelty of the approach and its good performance on multiple datasets. However, they also raised concerns about the limited applicability of the method to complex claims, the lack of discussions on integrating the approach into large language models, the efficiency of the pipeline, the scarcity of proof annotations on the test set, and the experimental fairness in the comparison with baselines.  The authors have provided a fair amount of discussion to the questions raised by the reviewers.